# A Clinical Perspective on Bespoke Sensing Mechanisms for Remote Monitoring and Rehabilitation of Neurological Diseases: Scoping Review

**DOI:** 10.3390/s23010536

**Published:** 2023-01-03

**Authors:** Jia Min Yen, Jeong Hoon Lim

**Affiliations:** 1Division of Rehabilitation Medicine, University Medicine Cluster, National University Hospital, Singapore 119074, Singapore; 2Department of Medicine, Yong Loo Lin School of Medicine, National University of Singapore, Singapore 119077, Singapore

**Keywords:** sensing mechanism, sensors, remote rehabilitation, remote monitoring, neurological disease, stroke, neurodegenerative disorder

## Abstract

Neurological diseases including stroke and neurodegenerative disorders cause a hefty burden on the healthcare system. Survivors experience significant impairment in mobility and daily activities, which requires extensive rehabilitative interventions to assist them to regain lost skills and restore independence. The advent of remote rehabilitation architecture and enabling technology mandates the elaboration of sensing mechanisms tailored to individual clinical needs. This study aims to review current trends in the application of sensing mechanisms in remote monitoring and rehabilitation in neurological diseases, and to provide clinical insights to develop bespoke sensing mechanisms. A systematic search was performed using the PubMED database to identify 16 papers published for the period between 2018 to 2022. Teleceptive sensors (56%) were utilized more often than wearable proximate sensors (50%). The most commonly used modality was infrared (38%) and acceleration force (38%), followed by RGB color, EMG, light and temperature, and radio signal. The strategy adopted to improve the sensing mechanism included a multimodal sensor, the application of multiple sensors, sensor fusion, and machine learning. Most of the stroke studies utilized biofeedback control systems (78%) while the majority of studies for neurodegenerative disorders used sensors for remote monitoring (57%). Functional assessment tools that the sensing mechanism may emulate to produce clinically valid information were proposed and factors affecting user adoption were described. Lastly, the limitations and directions for further development were discussed.

## 1. Introduction

Neurological diseases not only impact physical functioning but also affect the patient’s cognition and psychological health. The resultant disability restricts participation in premorbid social roles and entails an increased demand for care services. In the same context, the functional outcome of patients undergoing treatment for neurological diseases correlates with personal well-being as well as public welfare expenditures. Recently, the Global Burden of Disease reported that most neurological diseases led to a substantial increase in social burden from 1990 to 2017 [1]. Hence, delivering patient-centric rehabilitation interventions is integral to minimizing disability and maximizing independence.

The recent experience of pandemic-related service disruptions has accelerated the paradigm shift from conventional gym-based rehabilitation therapy to home-based therapy. Remote rehabilitation, an alternative mode of service delivery, has emerged to safeguard the continuity of care. It was reported that telerehabilitation was not inferior to in-person therapy to improve independence in activities of daily living (ADLs), balance, health-related quality of life, and depressive symptoms [2]. Ideally, this trend is bolstered by the integration of sensing mechanisms paired with processing algorithms for remote rehabilitation, which enables unmanned monitoring and quantifiable measurements. Given the aging population leading to an increased demand for rehabilitation coupled with the shortage of manpower, home-based remote therapy may be the mainstay of futuristic rehabilitation of neurological diseases.

There have been enormous technical advances and breakthrough innovations in designing sensors to detect user intent or behavior, composing algorithms to improve the accuracy of data interpretation or therapeutic efficacy, and creating human-machine interfaces combined with optimized interoperability between sensors and rehabilitative devices. Chen Y et al. performed a systemic review of home-based technologies for stroke rehabilitation including purposeful games, virtual reality, harnessing information and telecommunication technologies for telerehabilitation, robotic devices to augment or replace manual therapy, sensors, and mobile devices which connected users with sensors. The authors derived two main human factors in designing home-based technologies, which were engagement including motivation. and the home environment including understanding the social context [3]. However, the review did not elaborate on the sensor type or sensing modality. Alarcón-Aldana AC et al. focused on the therapeutic use of motion capture systems to aid post-stroke upper limb rehabilitation and the most commonly used were Kinect and inertial measurement units (IMUs) [4]. Spencer J et al. concluded that the evidence for biofeedback for post-stroke gait training was equivocal but showed promising effectiveness, which warranted better designed larger-scale studies [5]. Di Biase L et al. reviewed the various technologies used for gait analysis in Parkinson’s Disease (PD) but only few studies showed accurate algorithms that could be clinically useful for diagnosis and symptoms monitoring [6]. On the contrary, Ferreira-Sánchez MDR et al. concluded that the quantitative measurement of rigidity in PD was all valid and reliable using servomotors, inertial sensors, and biomechanical and neurophysiological study [7]. Aşuroğlu T et al. demonstrated that signals from ground reaction force (GRF) sensors analyzed with a deep learning approach, a combination of Convolutional Neural Networks (CNN) and Locally Weighted Random Forest (LWRF), were used successfully in disease detection and severity assessment of PD [8]. Açıcı K et al. provided a set for context awareness through wrist-worn sensors comprising accelerometers, magnetometers, and gyroscopes. The team presented a computational method for activity recognition and person identification from hand movements. They proved that multimodal sensors, such as accelerometers and magnetometers, can improve the accuracy of data compared with the individual use of each sensor type [9].

However, recent evidence suggests that physiological sensors can support remote assessment and management, but this is mainly observed in research settings and has not yet been translated to routine clinical care [10]. Considering a paucity of surveys scrutinizing the sensing mechanism utilized in clinical practice, especially such as remote monitoring and rehabilitation settings, it would be meaningful to collate published findings on how to generate suitable data required for tailored rehabilitative interventions according to the various clinical manifestations of neurological diseases. At the same time, it has to be underscored that the major challenges of remote rehabilitation would be to assure patient safety, adherence, and the efficacy of interventions. As clinicians who participate in the architecture development of remote rehabilitation and prescribe the system for end users, patients, we, the authors, hope to share opinions from a clinical point of view in this scoping review.

The aims of this study are to (1) review current trends in the application of sensing mechanisms in remote monitoring and rehabilitation with a focus on two broad categories of neurological diseases (stroke and neurodegenerative disorders (NDD)), and (2) to elaborate and propose the underpinnings to develop bespoke sensing mechanisms for remote rehabilitation from a clinical perspective.

## 2. Methods

### 2.1. Search Method

A systemic search of available literature in PubMed using Medical Subject Headings (MeSH) was performed following the PRISMA guidelines. Medical Subject Headings including “Stroke”, “Neurodegenerative Disease”, “Parkinson’s Disease”, “Alzheimer Disease”, “Dementia”, “Amyotrophic Lateral Sclerosis”, “Motor Neuron Disease”, with “Remote Sensing Technology” or “Telerehabilitation” was used.

The specific combinations that were used are:“Stroke” AND “Remote Sensing Technology”“Stroke” AND “Telerehabilitation”“Neurodegenerative Disease” AND “Remote Sensing Technology”“Neurodegenerative Disease” AND “Telerehabilitation”“Parkinson’s Disease” AND “Remote Sensing Technology”“Parkinson’s Disease” AND “Telerehabilitation”“Alzheimer Disease” AND “Remote Sensing Technology”“Alzheimer Disease” AND “Telerehabilitation”“Dementia” AND “Remote Sensing Technology”“Dementia” AND “Telerehabilitation”“Amyotrophic Lateral Sclerosis” AND “Remote Sensing Technology”“Amyotrophic Lateral Sclerosis” AND “Telerehabilitation”“Motor Neuron Disease” AND “Remote Sensing Technology”“Motor Neuron Disease” AND “Telerehabilitation”

### 2.2. Eligibility

We included English language articles of studies performed on human subjects published from 2018 to 2022. Editorials, reviews, and meta-analyses were excluded.

### 2.3. Selection of Study

PICO (population, intervention, comparison, outcome) principles were used for the selection criteria: (P) people with neurological diseases; (I) sensor usage in either remote monitoring or rehabilitation of neurological diseases; (C) none; (O) none. The search process is represented in Figure 1.

## 3. Results

### 3.1. Characteristics of Included Study

A total of sixteen studies were included in this review, of which nine studies involved stroke patients, and seven studies involved patients with NDD. Of these, three studies involved patients with dementia, and four studies involved patients with Parkinson’s Disease (PD).

### 3.2. Use of Sensors in Stroke

Nine studies describing the use of sensors for remote monitoring or rehabilitation were identified in stroke patients. Four studies utilized sensors in rehabilitation which resulted in improvement in upper limb function [11,12,13,14], Lee YM et al. used sensors to measure upper limb impairment and disability [15] and Song Y et al. used sensors for both stroke prevention and rehabilitation nursing via a mobile medical management system based on Internet-Of-Things technology [16]. Chen SC et al. demonstrated superior or equal efficiency of a Microsoft Kinect-based exergaming telerehabilitation system compared to conventional one-on-one physiotherapy in chronic stroke patients [17]. Salgueiro C et al. utilized a G-Walk accelerometer system to measure gait parameters in patients performing home-based core strengthening guided by a telerehabilitation application [18]. Rogerson L et al. also reported the feasibility of using The Howz system to identify activity abnormalities in stroke survivors [19]. Table 1 summarizes the use of sensors for remote monitoring or rehabilitation after stroke.

### 3.3. Use of Sensors in Neurodegenerative Disorders

Seven studies describing the use of sensors for remote monitoring or rehabilitation were identified in patients with NDD. Four studies involved patients with Parkinson’s disease, of which three studies used sensors to monitor parkinsonian manifestations [20,21,22] and Cikajlo I et al. described the use of Microsoft Kinect to calibrate the difficulty of games in a telerehabilitation exergaming system [23]. Another three studies involving patients with dementia were identified. Vahia IV et al. described the use of radio signal sensing and signal processing to identify behavioral symptoms of dementia [24]. Lazarou I et al. performed a randomized parallel trial, in which patients who received tailored non-pharmacological interventions according to observations by a sensor-based system showed statistically significant improvement in cognitive function [25]. However, Gaugler JE et al. concluded that ambient sensors placed to monitor daily activity did not affect caregiving outcomes over a 6-month follow-up period [26]. Table 2 summarizes the use of sensors for remote monitoring or rehabilitation in NDD.

### 3.4. Summary of Study Findings Regarding the Sensing Mechanism in Neurological Diseases

Figure 2 illustrates the statistical summary of all sixteen papers included in this review. As for publication years, one study was published in 2018 [23], three in 2019 [19,25,26], five in 2020 [11,14,20,21,24], four in 2021 [12,13,14,15], and three up till September 2022 [16,18,22]. The increased number of publications made since 2020 during the recent COVID-19 pandemic may reflect the growing interest in remote rehabilitation technology in neurological diseases. The most common study design was case series (50%), followed by a controlled clinical trial (25%), randomized parallel trial (13%), and case report (13%).

In our review, teleceptive sensors such as RGB cameras, depth sensors, infrared sensors, ambient sensors, and radio signal sensors (56%) [11,13,15,17,19,23,24,26] were utilized more frequently than wearable proximate sensors including accelerometer, gyroscope, pedometer, and surface EMG (44%) [12,14,16,18,20,21,22]. One study used both types of sensors—an ambient sensor and an accelerometer [25]. The sensing modality was infrared (38%) [11,13,15,17,23,25], acceleration force (38%) [16,18,20,21,22,25], RGB color (25%) [13,15,17,23], EMG (13%) [12,14], light and temperature (13%) [19,25], and radio signal (6%) [24], considering some studies used more than one sensing modality. The most common site of proximate sensor placement was the arm (38%) [12,14,16,21,22,25], followed by the trunk and leg (6%) [18]. The strategy adopted to elaborate the sensing mechanism was multimodal sensor (38%), application of multiple sensors (6%), sensor fusion (6%), and machine learning (6%) while the rest (43%) had no additional intervention.

As for the goals for adopting a remote sensing mechanism, there was a discrepancy between studies performed on stroke and NDD. Most of the stroke studies utilized a biofeedback control system (78%) [11,12,13,14,15,17,18] which aims to improve users’ limb function and balance. Of these, six studies (86%) [11,12,13,14,17,18] demonstrated the efficacy of a telerehabilitation strategy combined with the use of sensing mechanisms. While Marin-Pardo O et al. were unable to demonstrate normalization of co-contraction with EMG- based Tele-REINVENT, patients reported subjective improvements in motor function and quality of life [14]. In contrast, many studies in NDD used sensors for remote monitoring (57%) in order to provide diagnosis [20,22,24] and monitor pharmacological effects [21]. The studies using a biofeedback control system in NDD (43%) focused on improving cognitive function [25,26] and physical activity [23]. Of these, two studies (67%) showed statistically significant improvements in measured primary endpoints.

Of the sixteen studies, five (31%) reported technical issues [11,12,21,23,24]. Technical issues range from needing technical assistance with device [11,21,23], device failure [11], and Wi-Fi connectivity issues [24]. Six studies (38%) [11,12,13,17,19,24] reported no adverse events while the other ten studies (63%) did not report the rate of adverse events. Eleven out of the sixteen studies (69%) [11,12,13,16,17,21,22,23,24,25,26] kept datasets in a private repository while the remaining five (31%) [14,15,18,19,20] allowed public access. Eight out of the sixteen studies (50%) [15,17,18,19,21,23,24,26] adopted commercialized devices and the rest (50%) [11,12,13,14,16,20,22,25] utilized devices not yet commercially available.

## 4. Discussion

### 4.1. Clinical Considerations for an Ideal Sensing Mechanism in Remote Rehabilitation

The common purpose of applying sensors in neurological disease remains to facilitate remote rehabilitation regardless of diagnosis. The overarching principle of rehabilitation for neurological diseases would be to improve locomotive function and help conduct ADLs [27]. However, the specific goals for rehabilitative interventions should be tailored according to individual disease characteristics. In addition, all stakeholders participating in the development and application of sensors for remote rehabilitation need to consider the heterogenicity and complexity of neurological disorders. Though this paper divided neurological diseases into stroke and other neurodegenerative disorders to better understand the current practice, there would be inevitable overlaps in clinical features between disease groups [28,29]. In order to achieve the best outcomes of remote rehabilitation in these cases, a meticulous discussion to prioritize the goals should be held prior to adjusting the sensing mechanism based on the prevailing challenges. As such, rendering a suitable system for remote rehabilitation of neurological diseases would require a multifaceted approach.

#### 4.1.1. Commonly Used Clinical Parameters for Functional Assessment in Neurological Disease

A significant challenge remains when applying sensing mechanisms in the remote monitoring and rehabilitation of neurological diseases as classic sensor-based physiologic measures do not directly provide an index of a desired outcome measure [30]. However, the application of remote sensing mechanisms can provide objective measurements to derive close estimates for clinical assessments. Using sensors for continuous monitoring of patients’ motor function during their daily activities also furnishes complementary information to routine assessment tools [31].

That said, it would be important to understand what tools are used to evaluate the function in neurological diseases which sensors emulate by producing a sequence of data. In stroke, the commonly utilized measures include Fugl-Meyer Assessment (FMA) [32] for the locomotor function of upper and lower limbs, Functional Independence Measure (FIM) [33] or Barthel Index [34] for the ability to perform ADLs, Functional Ambulation Category [35] or Six-Minute Walk Test for the ability to walk, and Modified Tardieu Scale [36] or the Modified Ashworth Scale [37] for spasticity. In NDD, common clinical measures of PD include Movement Disorder Society-Unified Parkinson’s Disease Rating Scale (UPDRS) [38], gait speed, and Berg Balance Scale [39]. UPDRS values of patients are predicted with machine learning methods using wearable sensors aiming to provide prognosis solutions on rehabilitation areas [8,40] Dementia may cause changes in gait patterns such as decreased step and stride length, increased single limb stance time, double limb support time, and increased gait variability [41]. For dementia, assessment scales with established reliability include the Berg Balance Scale [42], Groningen meander walking test [43], Modified test for sensory interaction in balance [44], Step test [44], and Time up and go test [42].

#### 4.1.2. Elaboration of Sensing Mechanisms to Process Data Tailored to Clinical Needs

##### Multimodal Sensors

Multimodal sensors have been developed to detect various stimuli such as touch and proximity amalgamating discrete sensors [45]. Our review shows that this strategy is most commonly utilized by combining, for example, RGB cameras, depth sensors, and infrared sensors [23] or merging accelerometers, gyroscopes, and pedometers [20]. In addition, it was reported that the combination of IMUs with surface electromyography (sEMG) and mechanomyography was used to assess elbow spasticity [46]. Wrist-worn multimodal sensors comprising accelerometers, magnetometers, and gyroscopes improved the accuracy of data compared with the individual use of unimodal sensors [9]. Recently, to reduce the bulky size, a miniature multi-axial tactile sensor was fabricated by micro-electro-mechanical-system technology, which detects shear force using NiCr strain gauge film embedded in elastomer [47]. Developing multimodal sensors which can be incorporated easily into rehabilitation devices using nanotechnology would enhance the reliability of data and broaden the clinical applicability.

##### Applying Multiple Unimodal Sensors

Applying multiple unimodal sensors on different anatomical sites can generate stereotaxic data on top of the intrinsic biometric information. Salgueiro C et al. applied multiple accelerometers to the trunk and entire lower limb to measure gait parameters, in combination with a telerehabilitation application, and successfully demonstrated improvement of trunk function and sitting balance [18]. Oubre B et al. reported that two inertial sensors on the wrist and sternum measuring 3-dimensional random movements combined with unique movement decomposition techniques correlated moderately with upper limb FMA [48].

##### Sensor Fusion

Sensing modalities can be broadly classified into proximate versus teleception. Proximate sensing involves sensors that are wearable or in direct contact with the user. Examples include EMG sensors, load cells, linear encoders, smart fabric sensors, or IMU. In contrast, teleception or remote sensing is defined as sensing that occurs remotely, or with no physical contact being made with the object being sensed [49]. Teleceptive sensing may include sensors indirectly measuring the environment or behavior of things external to the user, such as RGB camera, IR sensor, laser/LED-based sensor, ultrasonic sensor, or Radar [50].

In our review, teleception is utilized more frequently (56%) than proximate sensing method (50%). The sum of teleception and proximate sensing is more than 100% because Lazarou I et al. adopted a sensor-fusion strategy by applying information from both ambient sensing (teleception) and proximate sensing (an accelerometer) to tailor non-pharmacological interventions in cognitive function, sleep quality, and daily activity [25]. From another clinical standpoint, both proximate sensing and teleceptive sensing mechanisms can provide kinetic and kinematic data for motion analysis. It can be highlighted that the former would be more effective in intent recognition by detecting subtle limb movement or neuromuscular activity, while the latter would be more useful to evaluate gait speed, balance, and gait pattern. Nonetheless, the fusion of proximate and teleceptive sensing mechanisms may provide a promising solution to tackle the challenges regarding the accuracy and clinical relevance of data acquired from sensors in the remote rehabilitation of neurological diseases.

##### Machine Learning Algorithms

To enable generalization in sequential data structures, enhance the accuracy of recognition, and achieve real-time forward prediction, the adoption of artificial intelligence technology, especially supervised machine learning algorithms, is instrumental in developing sensing mechanisms. As shown in Table 1 and Table 2, the machine learning algorithm has been underutilized as a sensing mechanism for remote rehabilitation settings. Song Y et al. applied backpropagation neural network for the assessment of arm function in stroke survivors, which showed prediction results of the mobile monitoring system for Brunnstrom stages I and II were completely consistent with the clinical staging results while in stages III-VI, the prediction accuracy was 90.63% [16]. The results demonstrate the pros and cons of backpropagation such as a simplified network structure useful to work on error-prone input data and sensitivity to noisy data. Abujrida H et al. applied Random Forest algorithm and captured features of PD gait anomalies through machine learning classification of smartphone sensor data collected in the home environment [20]. The team adopted two strategies, machine learning as well as multimodal sensors comprising an accelerometer, gyroscope, and pedometer, and showed that the stage and severity of PD can be inferred by machine learning classification of data acquired by multimodal smartphone sensors. Random Forest (RF) classifier [51,52] can be utilized in medical data analysis due to its ease of interpretation as well as its speed of learning for a big dataset. There are other machine learning classifier algorithms frequently used for sensing mechanisms in neurological diseases. Artificial Neural Networks (ANNs) which were inspired by the structure of neurons in the brain are mainly used for post-stroke rehabilitation assessments. Convolutional Neural Networks (CNN) [53], a division of ANNs, are used in the computer vision field with outstanding accuracy. Aşuroğlu T et al. introduced a supervised model, Locally Weighted Random Forest (LWRF) fed by ground reaction force signal and focused on predicting PD symptom severity to exploit relationships between gait signals [8]. Recently the same group demonstrated that a hybrid deep learning model, the combination of CNN and LWRF, outperformed most of the previous studies in disease detection and severity assessment of PD [40]. k-Nearest Neighbour (kNN) classifier [54] is a simple algorithm and is frequently used in real-time activity recognition. There have been trials to apply kNN to detect stroke [55] and heart disease [56]. However, the efficiency of the kNN algorithm is greatly reduced for large sample sizes and features. Cluster denoising and density cropping are suggested to improve efficiency [57]. Support Vector Machines (SVM) [58] are used for activity recognition and clinical assessments. Cai S et al. presented an upper-limb motion pattern recognition method using sEMG signals with SVM to conduct post-stroke upper-limb rehabilitation training [59]. Hamaguchi T et al. presented a non-linear SVM to analyze and validate finger kinematics using the leap motion controller and the outcome was compared with those assessed by therapists. The SVM-based classifier obtained high separation accuracy [60].

#### 4.1.3. Application of Feedback and Feedforward Control System to the Sensing Mechanism

In order to make remote rehabilitation a valid therapeutic alternative comparable to in-person gym rehabilitation, biometric data collected by sensors should be automatically linked to actuating architecture. Feedback and feedforward systems, a key element of industrial automation, could be applied for this purpose. A feedback system measures a specific variable and reacts when there is a shift, while a feedforward system may measure several variables simultaneously. Functional magnetic resonance imaging (fMRI) illustrates that different brain regions contribute to feedback and feedforward motor control processes and responds to global shifts in motor performance. Movements made to larger targets relied more on feedforward control whereas movements made to smaller targets relied more on feedback control [61].

For stroke rehabilitation, six out of the nine studies utilized sensors in the delivery of telerehabilitation. Of the six studies, three (50%) [11,14,17] incorporated both a feedforward and feedback control system into the sensing mechanism. The Home-based Virtual Rehabilitation System can feedforward motion signals into a cloud-based data server, then can feedback the data to calibrate the difficulty of rehabilitation games [11]. A bidirectional telerehabilitation exergaming system uses Microsoft Kinect to collect feedforward signals which are then transmitted to a database center to allow monitoring and feedback by a therapist remotely [17]. Similarly, Tele-REINVENT can feedforward EMG signals into a processing algorithm to enable feedback after offline analysis on top of immediate feedback provided by laptop recording, an occupational therapist, and real-time visualization of the EMG signals [14].

When composing a bespoke sensing mechanism with multimodal sensors or the sensor fusion approach, developers may adopt a feedforward system to process multiple inputs from the user and the environment. Given that feedforward systems cannot be accurate without an approximate process model, feedback controls should always be coupled with feedforward to provide a proper backup.

### 4.2. Factors Affecting the Adoption of Sensing Mechanism

A bespoke sensing mechanism for remote rehabilitation enhances the user experience of both the end user and the prescriber by optimizing user-technology-user interface. Based on the technology acceptance model by Davis in 1989, the “perceived ease of use” and “perceived usefulness” influence attitudes toward the usage of new technology [62]. For example, in an observational study, patients preferred wrist-worn sensors and those that provided the most effective feedback [63]. Possible other significant factors would include user trust, affordability, and practical features such as comfort and portability of the sensing mechanism. For instance, material selection for the development of smart fabric sensors would need to take into account breathability and stretchability. The virtual reality rehabilitation system (VRRS) demonstrated significantly better system usability compared to the Leap Motion Controller, which may be explained by the superior comfort of the VRRS [13]. Ensuring the user’s understanding of the sensing and control mechanism can build up user confidence and prevent fear of injuries or poor controllability [64]. The Howz system received positive reviews from study participants as users felt that sensors were nonintrusive, and their privacy was protected [19]. It may also improve patient satisfaction if clear instructional videos and robust technical support via video calls or emails can be provided [14].

For the prescriber, important performance prerequisites for sensing mechanisms are reliability and accuracy. Sensors used in remote rehabilitation need to be calibrated for noise created by movements during rehabilitation. Effects on measurements due to sensor disconnections and placement of leads also warrant consideration. Moreover, data also requires validation. With big data, which can be heterogeneous in nature, it will be important to parallel the development of data management capabilities and analyzing algorithms so that valuable information can be processed, interpreted, and translated into improvements in clinical practice. Importantly, protective measures will need to be in place to safeguard the confidentiality of data.

### 4.3. Limitations and Directions of Future Development

There are some limitations in this study. As literature searching was performed only on PubMED using Medical Subject Headings for pragmatic purposes, some relevant papers might not have been included. This review targeted research testing sensors specifically in remote monitoring or rehabilitation setting and removed many studies which were carried out offline or made online void of integration into remote rehabilitation systems. The value of this work is that the authors intended to share clinicians’ perspectives on the development of ideal sensing mechanisms for remote rehabilitation in neurological diseases rather than conduct a full-scale systemic review embracing issues about biomedical engineering and data science.

Remote rehabilitation for neurological diseases is burgeoning at the moment and may become mainstream in the near future due to its numerous merits. The key success factor of this innovative mode of service delivery would be to develop versatile smart sensors which can generate relevant clinical data and recognize user intent on a real-time basis. Extensive testing of the algorithms and functionality of sensors with many users in various ambiances would be required before a certain sensing mechanism is confirmed suitable for daily use. Machine learning algorithms should be included to minimize the trial-and-errors and expedite the development process.

Remote sensing mechanisms would move toward the integration of multiple sensors to achieve a target task. The fusion of different categories of sensors may maximize the synergistic output. For example, proximate sensors, such as inertial sensors, are incorporated with teleceptive sensors, such as IR sensors, to produce more accurate data about the target behavior. Multimodal sensors such as the combination of inertial data from IMUs and intrinsic muscle activity from sEMG enable dynamic motion analysis. Using multiple unimodal sensors can lower the risk of a system malfunction caused by a faulty sensor and can even obtain 3-dimensional data when purposefully placed at different body parts.

Though the inception of remote rehabilitation would be from the intent to reduce the burden associated with in-person consultation or gym therapy, a meticulously arranged discussion with healthcare professionals regarding the progress and updated goal setting would be crucial. To make this healthcare model feasible, an intuitive and user-friendly video consultation software network system should be installed in parallel. At the same time, an eHealth literacy program should be provided to the user according to sociodemographic factors affecting acceptance and readiness of the technology [65].

## 5. Conclusions

A variety of sensors are integrated into the architecture of remote rehabilitation for neurological diseases. The contemporary trend in the application of sensing mechanisms to stroke and NDD was described and the elements of functional assessment that sensors should emulate were discussed. The sensing mechanism can be further elaborated to generate purposefully processed information that can meet clinical standards by adopting multimodal sensors, sensor fusion, application of multiple sensors, and machine learning algorithms. The merits of feedback or feedforward control systems, the factors affecting the adoption of remote rehabilitation technology as end-user or prescribers, and the directions of future research were critically reviewed. Undeniably, there is a solid trend toward hybrid algorithms, multimodal sensing, sensor fusion, user comfort, and portability in sensor development for remote rehabilitation of neurological diseases. Precision remote rehabilitation in neurological disease can revolutionize the rehabilitation practice at the pace of the development of bespoke smart sensing mechanisms, which would require repeated testing and verification in a real-life environment.

## Figures and Tables

**Figure 1 sensors-23-00536-f001:**
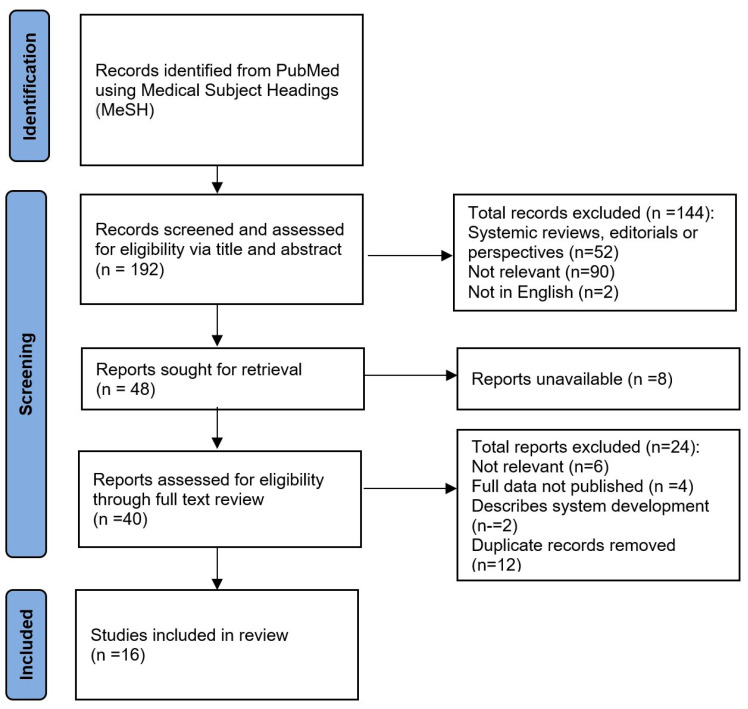
PRISMA flowchart summarizing the search process.

**Figure 2 sensors-23-00536-f002:**
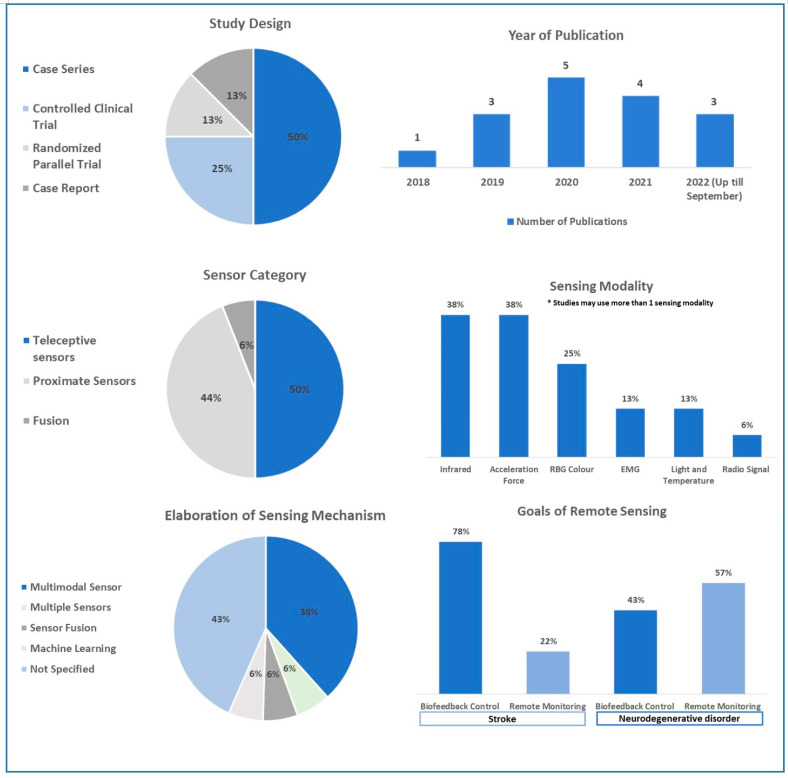
Statistics of Included Papers.

**Table 1 sensors-23-00536-t001:** Summary of identified studies involving stroke patients.

Authors, Published Year	Study Design	Study Sample	Device Name	Site	Sensor Type	Sensing Modality	Measured Clinical Parameters	Dataset (Repository)	Machine Learning Prediction	Results
Rogerson L et al., 2019 [19]	CS	19 stroke patients	Howz system (Commercialized)	NA	Ambient sensor, door sensor, and smart plug	Temperature, light	Mean number of times participant was active during the day, door sensor activation, alerts (due to low activity or late start)	Mean number of times participant was active during the day = 47.1 ± 55, Mean number of times door sensor was activated per day = 5 ± 2.4, number of alerts = 1.1 ± 1.2 (Public)	NA	Howz system for monitoring and feedback activities were feasible and acceptable for stroke survivors. No technological problems or adverse events were noted. The system was nonobtrusive, easy to use, and provided peace of mind that help would be at hand if needed.
Lee YM et al., 2020 [15]	CS	41 stroke patients	Microsoft Kinect (Commercialized)	NA	RGB camera, depth sensor, infrared sensor	RGB, depth, infrared	Upper extremity 3D Kinect-based reachable workspace, FMA, MI-UE, QuickDASH	Total upper limb FMA (n = 34) = 50.8 ± 19.5, MI-UE (n = 41) = 79.8 ± 20.1, QuickDASH = 32.5 ± 23.8. Correlation: total RSA and FMA total (R^2^ = 0.68, *p* < 0.01), total RSA and MI-UE (R^2^ = 0.65, *p* < 0.01), total RSA and QuickDASH (R^2^ = 0.42, *p* < 0.01) (Public)	NA	A Kinect-based reachable workspace could be a useful alternative outcome measure of upper limb impairment and disability. The total relative surface area of the paretic side correlated with FMA, MI, and QuickDASH scores.
Qiu Q et al., 2020 [11]	CS	15 stroke patients	LMC	NA	Infrared LEDs, infrared cameras	Infrared	Upper extremity FMA, hand kinematics (HOR, HOA, WPR, WPA, HRR, HRA)	Mean increase of upper extremity FMA = 5.2 (SE = 0.69, *p* < 0.001). Improved ROM: 15.83% for HOR, 27.50% for WPR, and 37.20% for HRR. Less error during tracing task (15.76% in HOA, 18.70% in WPA and 18.75% in HRA) (Private)	NA	HoVRS provides data for customizing upper limb rehabilitation in their home setting with minimal in person instruction or assistance. Improvements in upper limb function and six measurements of hand kinematics are noted with use of the system.
Chen SC et al., 2021 [17]	CCT	30 stroke patients (15 Kinect, 15 controls)	Microsoft Kinect (Commercialized)	NA	RGB camera, depth sensor, infrared sensor	RGB, depth, infrared	BBS, TUG, Modified Falls Efficacy Scale, MI, and FAC	Improvement in BBS in both groups (control group: *p* = 0.01, effect size = 0.49; experimental group: *p* = 0.01, effect size = 0.70). TUG scores in experimental group improved (*p* = 0.005, effect size = 0.70) (Private)	NA	Kinect-based interactive telerehabilitation system with remote therapist supervision has superior or equal efficiency compared to one-on-one physiotherapy. Compliance and safety of this interactive telerehabilitation system is observed.
Nam C et al., 2021 [12]	CS	11 stroke patients	NA	Wrist, hand	WH-ENMS	EMG	FMA, ARAT, WMFT, Motor FIM, MAS, EMG activation level, and the Co-contraction Index (CI) of the target muscles	Significant improvements (*p* < 0.05, paired sample *t* test) in the mean FMA full score (33.4 vs. 44.5), ARAT (19.3 vs. 26.7), WMFT score (39.2 vs. 45.9), WMFT time (51.6 vs. 45.7) before and after training. Significant decrease (*p* < 0.05, Wilcoxon’s signed rank test) in mean MAS scores at elbow (2.18 vs. 1.49), wrist (1.95 vs. 1.18) and finger (1.98 vs. 1.40) before and after training. Significant decreases in the EMG activation levels of the APB and FCR-FD and EMG co-contraction index of measured muscle pairs; and significant reductions in the number of movements and maximal trunk displacements. (Private)	NA	WH-ENMS-assisted home- based self-help rehabilitation was feasible and effective for improving upper limb function. Significantimprovements in the voluntary motor control and muscle coordination of the upper limb, increased smoothness and reduced compensatory trunk movement during arm reaching coordinated with distal movements, and release of muscular spasticity at the elbow, wrist, and fingers.
Cha K et al., 2021 [13]	RPT	27 stroke patients	VRRS, LMC	NA	RGB camera (VRRS), infrared LEDs, and cameras (LMC)	RGB, infrared	FMA, virtual body ownership, agency, location of the body, and usability	FMA pre- and post-training: Conventional therapy (23.44 vs. 28.11, *p* = 0.000, D-value = 4.67), LMC (27.67 vs. 33.56, *p* = 0.001, D-value = 5.89), VRSS (20.78 vs. 31.22, *p* = 0.000, D-value = 10.44). Significant difference (F = 5.426, *p* = 0.005) with a large effect size (ƞ2 = 0.361) in D-value between VRSS and conventional therapy. Significant difference (F = 5.426, *p* = 0.021) with a large effect size (ƞ2 = 0.221) in the D-value between VRSS and LMC. Significant differences between VRRS and LP in body ownership (3.2 vs. 1.3, *p* = 0.044), agency (6.8 vs. 4.2, *p* = 0.049), usability (69.7 vs. 55.8, *p* = 0.038) (Private)	NA	VRRS improved the users’ senses of body ownership, agency, and location of the body. Users preferred using the VRRS to using the LMC. VRRS promotes rehabilitation; FMA scores improved in all groups in experiment 2, the mean D-values of the FMA scores of the group using VRRS was significantly higher than the control groups.
Marin-Pardo O et al., 2021 [14]	CR	1 stroke patient	Tele-REINVE-NT	Forearm	EMG	Surface EMG	EMG signals, game performance, user experience, patient-reported change in motor function	No statistically significant changes for extensor (rho = 0.27, *p* = 0.164) or flexor (rho = −0.34, *p* = 0.071) muscle activity, game performance (rho = 0.29, *p* = 0.06). (Public)	NA	Muscle-computer interface system had no adverse events and patient did not perceive discomfort, pain, or fatigue. Normalization of co-contraction was not statistically significant. Patient reported positive changes in motor function and improved quality of life.
Song Y et al., 2022 [16]	CCT	32 stroke patients, 6 healthy control-s	Mobile medical managem-ent system based on IOT technology	Upper arm	Accelerometer	Acceleration force	Brunnstrom staging	When the noise intensity was 5%, 10%, 20%, 40%, and 60%, the MSE of the optimized median filtering algorithm were 54:17 ± 4:52, 103:52 ± 8:63, 215:42 ± 17:95, 1302:17 ± 108:51, and 4865:22 ± 455:26, respectively, and MSE of the median filtering algorithm before optimization were 2:17 ± 0:34, 15:41 ± 1:48, 21:52 ± 1:99, 52:42 ± 4:87, and 116:92 ± 8:63, respectively. PSNR of the optimized median filtering algorithm was significantly higher than that before optimization. Maximum prediction accuracy of 89.83% in the test set was achieved with 23 neurons. (Private)	BP neural network. Training time (2.5 s) and root MSE (0.29) of the model were lowest when Traingda was used. Training time and root MSE of traingda were significantly lower than traingd and traingdm functions. Training steps of traingda function were significantly different from those of traingd and traingdm functions. When transfer in the hidden layer and the input layer is tansig, the error percentage (7.56%) and root MSE (0.25) of model are minimum.	MSE of the signal showed a significant upward trend. Brunnstrom staging results were compared with the prediction results of the mobile monitoring system. Prediction results of Brunnstrom stages I and II were completely consistent with the clinical staging results. 3 samples or 9.37% showed different normal prediction results and clinical stage results in stages III–VI, and the prediction accuracy was 90.63%. There is certain application value for the rehabilitation of stroke patients.
Salgueiro C et al., 2022 [18]	CCT	30 stroke patients (15 G-Walk and 15 control-s)	G-Walk accelerom-eter system from BTS Bioengine-ering (Commercialized)	Trunk, entire lower limb	Accelerometer	Acceleration force	STIS2.0, S-FIST, S-PASS, BBS, the number of falls and gait parameters measured by the G-Walk accelerometer system	Improvements in S-TIS 2.0 balance pre- and post-intervention: control (4.27 vs. 4.31, *p* = 0.534), experimental group (4.73 vs. 6.71, *p* = 0.001), *p* = 0.007. Significant differences pre- and post-intervention intergroup S-TIS 2.0 total: control (7.33 vs. 7.46, *p* = 0.606), experimental group (7.60 vs. 10.36, *p* = 0.000), *p* = 0.032. BBS pre- and post-intervention, improved in both control (41.27 vs. 42.54, *p* = 0.009) and experimental groups (43.2 vs. 44.93, *p* = 0.029). (Public)	NA	The authors used an accelerometer to measure gait parameters. Performing core-strengthening exercises guided by a telerehabilitation application vs. conventional therapy seems to improve trunk function and sitting balance in chronic post-stroke.

**Legend:** APB—Abductor pollicis brevis; ARAT—Action Research Arm Test; BBS—Berg Balance Scale; BP—backpropagation; CCT—Controlled clinical trial; CR—Case report; CS—Case series; EMG—electromyography; FAC—Functional ambulation category; FCR-FD—flexor carpi radialis-flexor digitorum; FMA—Fugl-Meyer Assessment; HOA—Hand opening accuracy; HOR—Hand opening range; HRA—hand roll accuracy; HRR—Hand roll range; HoVRS—Home-based Virtual Rehabilitation System; IOT—Internet of Things: LED—light-emitting diode; LMC—Leap Motion Controller; MAS—Modified Ashworth Scale; MI-UE—Motricity Index for Upper Extremity; MSE—Mean square error; PSNR—Peak signal-to-noise ratio; QuickDASH—Disabilities of the Arm, Shoulder, and Hand; RFID—Radio frequency identification; RMSE—ROM—Range of motion; RPT—Randomized parallel trial; RSA—Relative surface area; SEM—Standard Error of the Mean; S-FIST—Spanish version of Function in Sitting Test; S-PASS—Spanish version of Postural Assessment Scale for Stroke Patients; STIS2.0—Spanish-Trunk Impairment Scale; TUG—Timed up and go; VRRS—Virtual reality rehabilitation system; WH-ENMS—Wrist-hand exoneuromusculoskeleton; WMFT—Wolf Motor Function Test; WPA—Wrist pitch accuracy; WPP—Wrist pitch range.

**Table 2 sensors-23-00536-t002:** Summary of identified studies involving patients with neurodegenerative disorders.

Authors/ Published Year	Study Design	Study Sample	Device Name	Site	Sensor Type	Sensing Modality	Measured Clinical Parameters	Dataset (Repository)	Machine Learning Prediction	Results
Cikajlo I et al., 2018 [23]	CS	26 patients with PD	Microsoft Kinect (Commercialized)	NA	RGB camera, depth sensor, infrared sensor	RGB, depth, infrared	Box and Blocks Test, UPDRS and daily activity Jebsen’s test, writing a letter, moving light objects, Nine-Hole Peg Test, PDQ-39	Statistically significant improvements in Box and Blocks Test (mean: 47 vs. 52, *p* = 0.002, Cohen’s d = 0.40), UPDRS III (mean: 27 vs. 29, *p* = 0.001, d = 0.22), and daily activity Jebsen’s test; writing a letter (mean: 24.0 vs. 20.6, *p* = 0.003, d = 0.23); and moving light objects (mean: 4.4 vs. 3.9, *p* = 0.006, d = 0.46). (Private)	NA	Telerehabilitation exergaming system which tracked participants’ movements and adapted the difficulty level of games in real-time is feasible but may require technical assistance. This resulted in clinically meaningful significant improvements. Nine-Hole Peg Test did not significantly improve. Participants claimed problems with mobility but less with ADLs and emotional well-being. (PDQ-39)
Gaugler JE et al., 2019 [26]	CCT	132 patients with AD or a related dementia	RAM System (GreatCall system) (Commercialized)	NA	Ambient sensor	Information not available	Qualitative outcomes of SSCQ, self-efficacy, burden, role captivity, role overload, and CES-D	At baseline and 6 months post RAM: SSCQ controls (24.26 vs. 23.73) treatment (24.17 vs. 23.33); Self-Efficacy controls (27.62 vs. 27.59) treatment (27.94 vs. 28.39); Burden: controls (37.01 vs. 40.93) treatment (37.59 vs. 40.40); Role Captivity: controls (6.35 vs. 6.56) treatment (6.13 vs. 6.74); Role Overload: controls (7.41 vs. 7.42) treatment (7.95 vs. 7.51); CES-D: controls (32.51 vs. 35.95) treatment (33.01 vs. 38.90). (Private)	NA	The system identifies significant behavioral changes by monitoring patterns in ADLs, generating an alert. Compared to controls without RAM, the RAM system did not significantly affect caregiving outcomes over a 6-month period. Themes of caregiver characteristics, care recipient characteristics and living arrangements were identified by qualitative analysis.
Lazarou I et al., 2019 [25]	RPT	18 patients (12 with MCI and 6 with AD)	Xtion Pro, Plugwise, Wireless Sensor Tag System, Presence sensors, Withings Aura, Jawbone	Wrist	Ambient sensor, accelerometer	Infrared, depth, temperature, humidity/moisture, light, pressure, acceleration force	Standard neuropsychological assessment, GDS, PSS, and NPI	Improvement in experimental group RAVLT total: M(SD) 38.67(13.53) to M(SD) = 45.83(15.94), *p* = 0.03). Significant difference in experimental group MMSE M(SD) = 28.33(1.86) compared to non-pharmacological interventions group M(SD) = 25.33(1.51) and regular care M(SD) = 25.17(2.79). Significant difference in RAVLT-learning between experimental group (M(SD) = 9.00(4.05) and non-pharmacological interventions group (M(SD) = 4.00(1.90). Significant difference in PSS of experimental group (M(SD) = 3.83(8.2) and regular care (M(SD) = 15.33(3.50). (Private)	NA	The experimental group received tailored non-pharmacological interventions according to system observations and showed improvement in the majority of neuropsychological tests (TEA, elevator time test, TRAIL-B, RBMT-recall, BDI) and statistically significant improvement in cognitive function, sleep quality, and daily activity compared to both control groups (tailored non-pharmacological interventions based on self-reported symptoms vs. neither system installation nor interventions)
Vahia IV et al., 2020 [24]	CR	1 patient with AD	The Emerald device (Commercialized)	NA	Radio signal sensor	Radio signals	Positional data, motion episodes (a segment of uninterrupted motion of ≥6 feet in one direction)	Mean motion episodes per day across all days = 82.7 (SD = 35.8). Significant (paired t test, *p* < 0.05) increase in motion episodes on days with family visits (93.8 (SD = 30.4) vs. non visit days 80.9 (SD = 36.3). Average 13.7% increase in motion episodes on visit days compared to the prior day and a 29.9% increase compared to the subsequent day. (Private)	NA	“The Emerald device” helps to identify behavioral symptoms of dementia on a day-to-day basis, while staff logs on patient behavior did not generate comparable temporally detailed information on behavior. The device transmitted 96.2% of data with no adverse events. Data may help identify and preempt triggers for BPSD.
Abujrida H et al., 2020 [20]	CS	152 patients with PD, 304 healthy controls	NA	NA	Smartphone sensor (accelerometer, gyroscope, pedometer)	Acceleration, angular velocity	Statistical, time, wavelet, and frequency domain features, and other lifestyle features	Gait features which decrease prediction error (MSE) in classification: (1) Entropy rate for walking balance severity (2) Lifestyle features and multiple gyroscope features for shaking/tremor, and (3) Accelerometer and gyroscope features for FoG Entropy rate and minMaxDiff (differences in step swing captured with accelerometer peaks) correlate linearly with gait severities. (Public)	Highest accuracy and AUC were (1) Random forest and entropy rate, 93% and 0.97, respectively, for walking balance; (2) Bagged trees and MinMaxDiff, 95% and 0.92 respectively, for shaking/tremor; (3) Bagged trees and entropy rate, 98% and 0.98 respectively, for FoG; and (4) Random forest and MinMaxDiff, 95% and 0.99 respectively, for distinguishing PD patients from HC False positive rate of classification is significantly higher if lifestyle features are not included.	Feature importance calculation based on machine learning is a better measure of feature significance Through machine learning classification of smartphone sensor data of PD gait anomalies collected in the home environment, the stage and severity of PD can be inferred.
Dominey T et al., 2020 [21]	CS	166 patients with PD	Parkinson’s KinetiGraph (PKG™) (Commercialized)	Wrist	Accelerometer	Acceleration force	Bradykinesia, dyskinesia, percentage of time with tremor, and percentage of time immobile	Most frequently reported findings in both follow-up and new patients were bradykinesia (63% and 72%, respectively) and sleep disturbance (58% and 41%, respectively). Treatment recommendations were made in 152/166 (92%) patients. Treatment recommendations were implemented for 83/114 (73%) patients, with advanced therapy in 6/9 (67%), additional motor agent in 34/71 (48%) and additional non-motor agent in 16/28 (57%). (Private)	NA	PKG™ indices with detection threshold for undertreatment were determined. The most common treatment changes relating to dopamine replacement and advice on sleep hygiene and bowel management. The study highlighted opportunities and challenges associated with incorporating digital data into care traditionally delivered via in-person contact.
Lipsmeier F et al., 2022 [22]	CS	316 subjects with PD	NA	Wrist	Smartphone/Smartwatch	Acceleration force	Bradykinesia, bradyphrenia and speech, tremor, gait, and balance	All pre-specified sensor features exhibited good-to-excellent test-retest reliability (median intraclass correlation coefficient = 0.9), and correlated with corresponding UPDRS items (rho: 0.12–0.71). Strongest correlations between sensor features and corresponding clinical items are observed with bradykinesia sensor features (Hand Turning and Finger Tapping), postural and rest tremor sensor features. Weakest correlations were found with the Balance and Draw A Shape tests. 15/17 sensor features discriminated participants with UPDRS scores of 0 vs. 1. 13/17 sensor features discriminated participants with H&Y stage I vs. II. (Private)	NA	The study demonstrated the preliminary reliability and validity of remote at-home quantification of motor sign severity with Roche PD Mobile application to assess motor signs in early PD and related movement disorders.

**Legend:** AD—Alzheimer’s disease; ADL—activity of daily living; AUC—Area under the curve; BDI—Beck Depression Inventory; BPSD—Behavioral and psychiatric symptoms of dementia; CCT—Controlled clinical trial; CES-D—Center for Epidemiological Studies—Depression; CR—Case report; CS—Case series; FoG—Freezing of gait; GDS—Global Deterioration Scale; H&Y—Hoehn and Yahr; ICC—Intraclass correlation coefficient; MSE—mean squared error; NPI—Neuropsychiatric Inventory; PD—Parkinson’s disease; PDQ-39—Parkinson’s Disease Questionnaire—39; PSS—Perceived Stress Scale; RAM—Remote activity monitoring; RAVLT—Rey Auditory Verbal Learning Test; RBMT—Rivermead Behavioral Memory Test; RPT—Randomized parallel trial; SSCQ—Short Sense of Competence Questionnaire; TEA—Test of Everyday Attention; UPDRS—Unified Parkinson Disease Rating Scale.

## Data Availability

No new data were created.

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
