# Peer review of "A Clinical Perspective on Bespoke Sensing Mechanisms for Remote Monitoring and Rehabilitation of Neurological Diseases: Scoping Review"

_sensors, 2023, doi:10.3390/s23010536_

Round 1

Reviewer 1 Report

Thanks for allowing me to evaluate your work titled "Bespoke sensing mechanisms for remote rehabilitation of neurological diseases: A clinical perspective" in which you perform a quasi-systematic review (no systematic data extraction) about remote sensing mechanisms in stroke and neurodegenerative disorders. 

I think the topic is timely, of interest, and can surely help advance the field. 

The main aspect that could be improved is to expand the results and perform more systematic data extraction to enhance the systematic nature of the search. Also, connecting these results with the discussion. Currently, it gave me the impression that the results and the discussion are not well linked.

Some additional specific comments in case they are helpful: 

1. It might be good to add "systematic review" to the title, especially if the data extraction is enhanced.

2. In your search some of the included studies (e.g., Lispmeier et Al, Vahia et al, and there might be others)  do not necessarily focus on telerehabilitation but rather on remote sensing. Please consider changing the title, and the objectives if you decide to include these articles. 

3. In table 1, I will suggest changing the "Subject" column title to population or "Study Sample".

4. Please check the word symptoms throughout the text. In section 3.3 this seems to refer more to signs. To avoid this type of semantic issue, it might be better to use manifestations instead of symptoms.

5. As I said, the results section is light and disconnected from the discussion. I will move section 4.1 and Fig 2. to the results section. I will suggest only including in 4.1 a more concise summary of the main findings. I think that you could also expand the results section with some systematic extraction of results from the paper and for example add (i) the outcomes that were assessed, (ii) if the telerehabilitation strategy showed efficacy or met the primary endpoint, (iii) also if there were any adverse outcomes reported, or (iv) technical issues, (v) if the system has been translated/commercialized/approved by regulators, (vi) and also the geographical location of the initiative?

6. Section 4.2.2 was difficult to follow as the individual paragraphs were not always connected. It might help to break down this section by adding subheadings to the different paragraphs discussing different aspects.

7. The section where you discuss Machine Learning sounded a little bit speculative to me and it might be better to include some systematic data extraction for this aspect as well in the results and comment here on some of these results to back up your statements. Same for the 4.2.3 section of feedback and freedforward that could benefit from some actual assessment of these strategies in the cited works.

8. It will be beneficial to define somewhere (methods) what teleceptive and proximate sensors mean explicitly. 

Author Response

Dear Reviewer

We appreciate your insightful comments and advice to improve the paper and provide a point-by-point response. 

  1. It might be good to add "systematic review" to the title, especially if the data extraction is enhanced.

A) The value of this work is that the authors intended to share clinicians’ perspectives on the development of sensing mechanisms for remote rehabilitation in neurological diseases rather than conduct a full-scale systemic review embracing issues about biomedical engineering and data science. This review targeted research testing sensors specifically in remote monitoring or rehabilitation setting and removed many studies which were carried out offline or made online void of integration into remote rehabilitation systems. As such the authors would rather amend the title to define the character of this study, quasi-systematic review, by adding a "scoping review".

Amended title: A clinical perspective on bespoke sensing mechanisms for remote monitoring and rehabilitation of neurological diseases: Scoping review

  1. In your search some of the included studies (e.g., Lispmeier et Al, Vahia et al, and there might be others) do not necessarily focus on telerehabilitation but rather on remote sensing. Please consider changing the title, and the objectives if you decide to include these articles.

A) Monitoring the manifestations of the disease is crucial to make a clinical decision by detecting any changes in condition in time, which mandates remote sensing to be included in telerehabilitation. The scope of telerehabilitation comprises two pillars, remote monitoring (or sensing) and biofeedback (or feedforward) control system, which enable interactive rehabilitative intervention. To clarify the objectives, the authors added the term “remote sensing” to “remote rehabilitation” though the latter embraces the former.  

Amended objectives:

The aims of this study are to 1) review current trends in the application of sensing mechanisms in remote monitoring and rehabilitation with a focus on two broad categories of neurological diseases (stroke and neurodegenerative disorders (NDD)), and 2) to elaborate and propose the underpinnings to develop bespoke sensing mechanisms for remote rehabilitation from a clinical perspective.

  1. In table 1, I will suggest changing the "Subject" column title to population or "Study Sample".

A) The column title was amended accordingly.

  1. Please check the word symptoms throughout the text. In section 3.3 this seems to refer more to signs. To avoid this type of semantic issue, it might be better to use manifestations instead of symptoms.

A) The terminology was amended as suggested.

Amended: Four studies involved patients with Parkinson’s disease, of which three studies used sensors to monitor parkinsonian manifestations.

  1. As I said, the results section is light and disconnected from the discussion. I will move section 4.1 and Fig 2. to the results section. I will suggest only including in 4.1 a more concise summary of the main findings. I think that you could also expand the results section with some systematic extraction of results from the paper and for example add (i) the outcomes that were assessed, (ii) if the telerehabilitation strategy showed efficacy or met the primary endpoint, (iii) also if there were any adverse outcomes reported, or (iv) technical issues, (v) if the system has been translated/commercialized/approved by regulators, (vi) and also the geographical location of the initiative?

A) According to the suggestion, subsection 4.1 and Figure 2 were moved to the results section. The results section which consists of tables, figures, and descriptive illustrations was expanded to include (i) the outcomes that were assessed, (ii) if the telerehabilitation strategy showed efficacy or met the primary endpoint, (iii) also if there were any adverse outcomes reported, or (iv) technical issues, and (v) if the system has been translated/commercialized/approved by regulators. By the way, given the multinational collaboration of researchers in most of the papers, the authors concluded that “(vi) the geographical location of the initiative” might be less meaningful. The connection between the results section and the discussion section was reinforced by citing the works of the results section in the discussion.

  1. Section 4.2.2 was difficult to follow as the individual paragraphs were not always connected. It might help to break down this section by adding subheadings to the different paragraphs discussing different aspects.

A) The subheadings were added as suggested.

4.1.2. Elaboration of sensing mechanism to process data tailored to clinical need  

4.1.2.1. Multimodal sensors

4.1.2.2. Applying multiple unimodal sensors

4.1.2.3. Sensor fusion

4.1.2.4. Machine learning algorithms

  1. The section where you discuss Machine Learning sounded a little bit speculative to me and it might be better to include some systematic data extraction for this aspect as well in the results and comment here on some of these results to back up your statements. Same for the 4.2.3 section of feedback and feedforward that could benefit from some actual assessment of these strategies in the cited works.

A) The sections on machine learning and feedback-feedforward were bolstered by systematic data extraction for respective aspects and closely linked to cited papers by critically analyzing the application examples in each work as follows:

To enable generalization in sequential data structures, enhance the accuracy of recognition and achieve real-time forward prediction, the adoption of artificial intelligence technology, especially supervised machine learning algorithms, is instrumental in developing sensing mechanisms. As shown in the Table 1 and 2, machine learning algorithm has been underutilized in sensing mechanism for remote rehabilitation settings. Song Y et al applied backpropagation neural network for the assessment of arm function in stroke survivors, which showed prediction results of the mobile monitoring system for Brunnstrom stages I and II were completely consistent with the clinical staging results while in stages III-VI, the prediction accuracy was 90.63%. The results demonstrate the pros and cons of backpropagation such as a simplified network structure useful to work on error-prone input data and sensitivity to noisy data. Abujrida H et al applied Random Forest algorithm and captured features of PD gait anomalies through machine learning classification of smartphone sensor data collected in the home environment. The team adopted two strategies, machine learning as well as multimodal sensors comprising an accelerometer, gyroscope, and pedometer, and showed that the stage and severity of PD can be inferred by machine learning classification of data acquired by multimodal smartphone sensors. Random Forest (RF) classifier can be utilized in medical data analysis due to its ease of interpretation as well as its speed of learning for a big dataset. There are other machine learning classifier algorithms frequently used for sensing mechanisms in neurological diseases. Artificial Neural Networks (ANNs) which were inspired by the structure of neurons in the brain are mainly used for post-stroke rehabilitation assessments. Convolutional Neural Networks (CNN), a division of ANNs, are used in the computer vision field with outstanding accuracy. Aşuroğlu T et al introduced a supervised model, Locally Weighted Random Forest (LWRF) fed by ground reaction force signal and focused on predicting PD symptom severity to exploit relationships between gait signals. Recently the same group demonstrated that a hybrid deep learning model, the combination of CNN and LWRF, outperformed most of the previous studies in disease detection and severity assessment of PD. k-Nearest Neighbour (kNN) classifier is a simple algorithm and is frequently used in real-time activity recognition. There have been trials to apply kNN to detect stroke and heart disease. However, the efficiency of the kNN algorithm is greatly reduced for large sample sizes and features. Cluster denoising and density cropping are suggested to improve efficiency. Support Vector Machines (SVM) are used for activity recognition and clinical assessments. Cai S et al presented an upper-limb motion pattern recognition method using sEMG signals with SVM to conduct post-stroke upper-limb rehabilitation training. Hamaguchi T et al presented a non-linear SVM to analyze and validate finger kinematics using the leap motion controller and the outcome was compared with those assessed by therapists. The SVM-based classifier obtained high separation accuracy.

  1. It will be beneficial to define somewhere (methods) what teleceptive and proximate sensors mean explicitly.

A) The concept of teleceptive and proximate sensors was defined explicitly in subsection 4.1.2.3. as follows:

Sensing modalities can be broadly classified into proximate versus teleception. Proximate sensing involves sensors that are wearable or in direct contact with the user.  Examples include EMG sensors, load cells, linear encoders, smart fabric sensors, or IMU. In contrast, teleception or remote sensing is defined as sensing that occurs remotely, or with no physical contact being made with the object being sensed. Teleceptive sensing may include sensors indirectly measuring the environment or behavior of things external to the user, such as RGB camera, IR sensor, laser/LED-based sensor, ultrasonic sensor, or Radar.  

Thank you.

Reviewer 2 Report

Authors conducted a review on remote rehabilitation of neurological diseases. I have the following suggestions:

·         Are there any similar work done on this area? If so, authors should write a related works section.

·         In my opinion, 16 papers is not sufficient to conduct a review. Including more journal databases will increase this number and validity of the review.

·         References of these 16 papers can also be analyzed.

·         Why authors just used PubMed database? The authors should expand their search on databases like ScienceDirect and IEEE Xplore.

·        Remote sensing technology search term may not be the best term because it overlaps with Remote sensing which is a subset Geographical Information Systems.

·        Following studies can also be included in Parkinson’s Disease monitoring category. In these studies, UPDRS values of patients are predicted with machine learning methods using wearable sensors aiming to provide prognosis solutions on rehabilitation areas.

Parkinson's disease monitoring from gait analysis via foot-worn sensors, 2018.

 A deep learning approach for Parkinson’s disease severity assessment, 2022.

·         What about machine learning prediction results in table 1 and 2?  Authors should also report these results.

·         For table 1 and 2, dataset information should be added. For example, private dataset or public repositories.

·         Although authors conduct this review with clinical perspective important machine learning studies can also be added to increase contribution of this paper.

Author Response

Dear Reviewer

We appreciate your insightful comments and advice to improve the paper and provide a point-by-point response. 

  • Are there any similar work done on this area? If so, authors should write a related works section.

A) According to the reviewer’s advice, the related works were summarized at the end of the introduction section as follows:

There have been enormous technical advances and breakthrough innovations in designing sensors to detect user intent or behavior, composing algorithms to improve the accuracy of data interpretation or therapeutic efficacy, and creating human-machine interfaces combined with optimized interoperability between sensors and rehabilitative devices. Chen Y et al performed a systemic review of home-based technologies for stroke rehabilitation including purposeful games, virtual reality, harnessing information and telecommunication technologies for telerehabilitation, robotic devices to augment or replace manual therapy, sensors, and mobile devices which connect users with sensors. The authors concluded that the two main human factors in designing home-based technologies for stroke rehabilitation are designing for engagement and for the home environment. However, this review did not elaborate on the sensor type or sensing modality. Alarcón-Aldana AC et al focused on the therapeutic use of motion capture systems to aid upper limb rehabilitation, the most commonly used are Kinect and of inertial measurement units (IMUs). Spencer J et al concluded that the evidence for biofeedback for post-stroke gait training is equivocal but shows promising effectiveness which better designed larger-scale studies should further research. Di Biase L et al reviewed the various technologies used for gait analysis in Parkinson’s Disease (PD) but only few studies showed accurate algorithms that can potentially be clinically useful for diagnosis and symptoms monitoring. On the contrary, Ferreira-Sánchez MDR et al concluded that the quantitative measurement of rigidity in PD using servomotors, inertial sensors, and biomechanical and neurophysiological study of muscles all are valid and reliable. Aşuroğlu T et al demonstrated that signals from ground reaction force (GRF) sensors analyzed with a deep learning approach, a combination of Convolutional Neural Networks and Locally Weighted Random Forest (LWRF), can be used successfully in disease detection and severity assessment of PD. Açıcı K et al provided a set for context-awareness through wrist-worn sensors comprising accelerometers, magnetometers, and gyroscopes. The team presented a computational method for activity recognition and person identification from hand movements. Multimodal sensors, such as accelerometers and magnetometers, were proven to improve the accuracy of data compared with the individual use of each sensor type.

  • In my opinion, 16 papers is not sufficient to conduct a review. Including more journal databases will increase this number and validity of the review.
  • References of these 16 papers can also be analyzed.
  • Why authors just used PubMed database? The authors should expand their search on databases like ScienceDirect and IEEE Xplore.

A) The answer to the above 3 questions is as follows:

We totally agree with the reviewer’s suggestion to include more journal databases.  By the way, we seek the reviewer’s understanding of the situation that “Sensors” editorial office confirmed this topic for the special issue on 10th October 2022 only with fixing the submission deadline to 20th November 2022. Given the extremely tight timeline, to make this work feasible, the authors decided to provide a clinician’s perspective on the development of sensing mechanisms for remote rehabilitation in the form of a scoping review using PUBMED only instead of a full-scale systemic review embracing all available journal databases. The authors cited relevant references from the selected 16 papers to supplement the analysis and discuss the implications of the findings.   

  • Remote sensing technology search term may not be the best term because it overlaps with Remote sensing which is a subset Geographical Information Systems.

A) The title and the keywords of this paper were discussed with the editorial office to be compatible with the overarching theme of the Special Issue "Wearable Sensors for Neurological Diseases Remote Monitoring". As the reviewer expected, we encountered some GIS-related works but excluded them meticulously.

  • Following studies can also be included in Parkinson’s Disease monitoring category. In these studies, UPDRS values of patients are predicted with machine learning methods using wearable sensors aiming to provide prognosis solutions on rehabilitation areas.

Parkinson's disease monitoring from gait analysis via foot-worn sensors, 2018.

A deep learning approach for Parkinson’s disease severity assessment, 2022.

A) We appreciate the reviewer’s recommendation of the brilliant works. The two papers were cited in multiple sections of this paper to introduce related works, exemplify the application of algorithms for clinical assessment, and discuss machine learning as follows: 

(Introduction)

(Discussion)

4.1.1. Commonly used clinical parameters

4.1.2.4. Machine learning algorithms

  • What about machine learning prediction results in table 1 and 2?  Authors should also report these results.

A) The results were tabulated in Table 1 and 2 and discussed in detail in the machine learning subsection also as follows:

 As shown in the Table 1 and 2, machine learning algorithm has been underutilized in sensing mechanism for remote rehabilitation settings. Song Y et al applied backpropagation neural network for the assessment of arm function in stroke survivors, which showed prediction results of the mobile monitoring system for Brunnstrom stages I and II were completely consistent with the clinical staging results while in stages III-VI, the prediction accuracy was 90.63%. The results demonstrate the pros and cons of backpropagation such as a simplified network structure useful to work on error-prone input data and sensitivity to noisy data. Abujrida H et al applied Random Forest algorithm and captured features of PD gait anomalies through machine learning classification of smartphone sensor data collected in the home environment. The team adopted two strategies, machine learning as well as multimodal sensors comprising an accelerometer, gyroscope, and pedometer, and showed that the stage and severity of PD can be inferred by machine learning classification of data acquired by multimodal smartphone sensors.

  • For table 1 and 2, dataset information should be added. For example, private dataset or public repositories.

A) The results were tabulated in Table 1 and 2 and discussed in detail in the results section as follows:

Eleven out of the sixteen studies (69%) [11,12,13,16,17,21,22,23,24,25,26] kept datasets in a private repository while the remaining five (31%) [14,15,18,19,20] allowed public access to the datasets.

  • Although authors conduct this review with clinical perspective important machine learning studies can also be added to increase contribution of this paper.

A) The subsection of machine learning was bolstered by systematic data extraction for respective aspects and closely linked to cited papers by critically analyzing the application examples in each work as follows:

To enable generalization in sequential data structures, enhance the accuracy of recognition and achieve real-time forward prediction, the adoption of artificial intelligence technology, especially supervised machine learning algorithms, is instrumental in developing sensing mechanisms. As shown in Table 1 and 2, machine learning algorithm has been underutilized in sensing mechanism for remote rehabilitation settings. Song Y et al applied backpropagation neural network for the assessment of arm function in stroke survivors, which showed prediction results of the mobile monitoring system for Brunnstrom stages I and II were completely consistent with the clinical staging results while in stages III-VI, the prediction accuracy was 90.63%. The results demonstrate the pros and cons of backpropagation such as a simplified network structure useful to work on error-prone input data and sensitivity to noisy data. Abujrida H et al applied Random Forest algorithm and captured features of PD gait anomalies through machine learning classification of smartphone sensor data collected in the home environment. The team adopted two strategies, machine learning as well as multimodal sensors comprising an accelerometer, gyroscope, and pedometer, and showed that the stage and severity of PD can be inferred by machine learning classification of data acquired by multimodal smartphone sensors. Random Forest (RF) classifier can be utilized in medical data analysis due to its ease of interpretation as well as its speed of learning for a big dataset. There are other machine learning classifier algorithms frequently used for sensing mechanisms in neurological diseases. s. Artificial Neural Networks (ANNs) which were inspired by the structure of neurons in the brain are mainly used for post-stroke rehabilitation assessments. Convolutional Neural Networks (CNN), a division of ANNs, are used in the computer vision field with outstanding accuracy. Aşuroğlu T et al introduced a supervised model, Locally Weighted Random Forest (LWRF) fed by ground reaction force signal and focused on predicting PD symptom severity to exploit relationships between gait signals. Recently the same group demonstrated that a hybrid deep learning model, the combination of CNN and LWRF, outperformed most of the previous studies in disease detection and severity assessment of PD. k-Nearest Neighbour (kNN) classifier is a simple algorithm and is frequently used in real-time activity recognition. There have been trials to apply kNN to detect stroke and heart disease. However, the efficiency of the kNN algorithm is greatly reduced for large sample sizes and features. Cluster denoising and density cropping are suggested to improve efficiency. Support Vector Machines (SVM) are used for activity recognition and clinical assessments. Cai S et al presented an upper-limb motion pattern recognition method using sEMG signals with SVM to conduct post-stroke upper-limb rehabilitation training. Hamaguchi T et al presented a non-linear SVM to analyze and validate finger kinematics using the leap motion controller and the outcome was compared with those assessed by therapists. The SVM-based classifier obtained high separation accuracy.

Thank you. 

Reviewer 3 Report

The article reviews available methods for remote sensing and rehabilitation of neurological diseases. The presented review and analysis methodology is sound, however, several worthwhile projects can be added except the selected 16 in the study. The analysis (tables) can be supplemented with information on utilized biomedical data channels, detection algorithms (utilized signal analysis methods), and identified disease symptoms. 

Such a paper could recommend the most valuable solutions in terms of remote monitoring and assistance (treatment). Mainly the research should elaborate on the most valuable methods (algorithms) for biomedical sensor data analysis. There is a paragraph that discusses this issue but to provide valuable insights, authors should consider discussing the effectiveness of methods. 

The conclusion should contain recommendations and an assessment of the availability of tools and, their application perspectives, especially in the domain of neurological symptoms monitoring accuracy, mobility of measurements, the duration of the measurement process, usability of provided monitoring tools, and their simplicity, ease of use, ergonomics. 

Language is clear.

Author Response

Dear Reviewer

We appreciate your insightful comments and advice to improve the paper and provide a point-by-point response. 

The article reviews available methods for remote sensing and rehabilitation of neurological diseases. The presented review and analysis methodology is sound, however, several worthwhile projects can be added except the selected 16 in the study. The analysis (tables) can be supplemented with information on utilized biomedical data channels, detection algorithms (utilized signal analysis methods), and identified disease symptoms.

A) We hope the reviewer understand that this review targeted research testing sensors specifically in remote monitoring or rehabilitation setting and removed many studies which were carried out offline or made online void of integration into remote rehabilitation systems. Though not included in this study, several worthwhile projects and related works were summarized in the introduction section. Complying with the reviewer’s suggestion, Table 1 and 2 were revised to include information on utilized biomedical data channels, detection algorithms (utilized signal analysis methods), and identified disease symptoms.

Such a paper could recommend the most valuable solutions in terms of remote monitoring and assistance (treatment). Mainly the research should elaborate on the most valuable methods (algorithms) for biomedical sensor data analysis. There is a paragraph that discusses this issue but to provide valuable insights, authors should consider discussing the effectiveness of methods.

A) It would be difficult to recommend the most valuable solution or algorithm which can be uniformly applicable to remote monitoring or rehabilitation given the various disease behavior, clinical course and patient's need. Instead, the authors tried to provide a perspective as clinician practitioners on how to customize sensing mechanisms for rehabilitative interventions according to the various clinical manifestations and how to enhance user adoption. As per the reviewer’s suggestion, we revised the paper to further detail the strategy to elaborate sensing mechanisms such as multimodal sensors, applying multiple unimodal sensors, sensor fusion, and machine learning algorithms discussing the effectiveness of each method. We supplemented the contents on feedback and feedforward control system and analyzed the applications and the effectiveness in the cited works.

The conclusion should contain recommendations and an assessment of the availability of tools and, their application perspectives, especially in the domain of neurological symptoms monitoring accuracy, mobility of measurements, the duration of the measurement process, usability of provided monitoring tools, and their simplicity, ease of use, ergonomics.

A) Though it would be hard to specify all information in the conclusion as per the reviewer’s suggestion, the authors inserted succinctly the current trend toward future development of sensing mechanisms in remote rehabilitation as follows:

A variety of sensors are integrated into the architecture of remote rehabilitation for neurological diseases. The contemporary trend in the application of sensing mechanisms to stroke and NDD was described and the elements of functional assessment that sensors should emulate were discussed. The sensing mechanism can be further elaborated to generate purposefully processed information that can meet clinical standards by adopting multimodal sensors, sensor fusion, application of multiple sensors, and machine learning algorithms. The merits of feedback or feedforward control systems, the factors affecting the adoption of remote rehabilitation technology as end-user or prescribers, and the directions of future research were critically reviewed. Undeniably, there is a solid trend toward hybrid algorithms, multimodal sensing, sensor fusion, user comfort, and portability in sensor development for remote rehabilitation of neurological diseases. Precision remote rehabilitation in neurological disease can revolutionize the rehabilitation practice at the pace of the development of bespoke smart sensing mechanisms, which would require repeated testing and verification in a real-life environment.

Thank you. 

Thank you. 

Round 2

Reviewer 1 Report

Thanks for addressing my comments. 

Reviewer 2 Report

Thank you for submitting your revised manuscript. The manuscript has been significantly improved since the first submission and I'm overall satisfied with the corrections